# Effects of Maillard Reaction on Volatile Compounds and Antioxidant Capacity of Cat Food Attractant

**DOI:** 10.3390/molecules27217239

**Published:** 2022-10-25

**Authors:** Kekui Sun, Zhaoqi Dai, Wenlong Hong, Jianying Zhao, Hang Zhao, Ji Luo, Guangjie Xie

**Affiliations:** 1College of Tourism, Huangshan University, Huangshan 245041, China; 2Department of Tea and Food Science and Technology, Jiangsu Vocational College of Agriculture and Forestry, Jurong 212400, China; 3College of Life Science, Anhui Normal University, Wuhu 241000, China; 4Institute of Agricultural Products Processing, Jiangsu Academy of Agricultural Sciences, Nanjing 210014, China; 5Zhenjiang Zhinong Food Limited Company, Zhenjiang 212000, China; 6College of Food Science and Technology, Nanjing Agricultural University, Nanjing 210095, China

**Keywords:** Maillard reaction, cat food attractant, volatile compounds, antioxidant activity, palatability test

## Abstract

In this study, self-made cat food attractant was prepared by Maillard reaction using hydrolysate of grass carp waste as raw material and glucose and cysteine hydrochloride as substrate. Its volatile compounds, antioxidant capacity, and pet palatability were investigated. The volatile compounds of attractants were analyzed using gas chromatography–mass spectrometry (GC-MS) which showed that alcohols and aldehydes were the most volatile in self-made attractants, accounting for 34.29% and 33.52%, respectively. Furthermore, Maillard reaction could significantly increase the antioxidant activity of self-made attractant, including scavenging activity on OH and DPPH free radicals as well as the chelating ability of Fe^2+^. The acceptance and palatability of two kinds of cat food were studied by adding 3% self-made or commercial attractants. The results of this study also found that both attractants could remarkably improve the intake rate of cat food. However, the self-made group was significantly less than the commercial group in first smell, first bite, and feeding rate, which might be because of the absence of umami ingredients and spices in self-made attractants.

## 1. Introduction

Grass carp (*Ctenopharyngodon idellus*) is one of the four fish in a freshwater culture with rich nutrition and high protein content [1]. Fish waste refers to the waste produced in fish processing, such as fish viscera, fish skin, fish scale, fish head, fish bone, and fishtail, which is a rich protein resource [2]. At present, the disposal of fish waste material is mainly discarded and buried, which wastes resources and pollutes the environment around the world [3]. In the last few decades, discarded fish waste materials have attracted increasingly more attention, prompting researchers to develop them into valuable products, such as animal feed, natural pigment, soil fertilizer, packaging material for food, and some enzymatic products [4,5].

Maillard reaction refers mainly to the complex reaction between reducing sugars, amino acids, and proteins [6] and can be divided into three stages: primary reaction, intermediate reaction, and final reaction. The primary reaction stage is the formation of flavor precursors, including carbonyl ammonia condensation and molecular rearrangement; the intermediate reaction stage is essential for the formation of volatile substances, mainly the further degradation of the initial products. However, the final stage reaction is quite complex and unclear [7]. The main factors affecting the Maillard reaction are amino acids and carbohydrates; for example, cysteine was a vital ingredient in meat aroma, while phenylalanine could produce a unique violet aroma in meat products [8]. Since monosaccharide reaction products have better flavor than disaccharide reaction products, monosaccharide is consistently chosen as the reaction substrate; glucose and xylose are often used as raw materials for Maillard reaction because of their low price and high reactivity [9].

Maillard reaction is an important source of flavor and color in food processing, especially in the production of condiments. It was found that Maillard reaction products (MRPs) could not only produce a unique aroma but also have specific antioxidant effects [10]. Hodge and Rist [11] found that MRPs could inhibit the oxidation of vegetable oils and slow down the rate of spoilage. MRPs are self-produced substances without added chemicals; thus, they are natural and harmless and could replace antioxidants used in the food industry. Some experts have advocated adding MRPs to the food system or making the food itself from MRPs to improve the antioxidant stability of the product [12,13]. Chen et al. [14] expounded that the reaction of heated cod skin collagen peptides and xylose mixture had a higher browning level and more potent antioxidant activity than directly heated cod skin collagen peptides. With the extension of reaction temperature, the antioxidant activity of MRPs was stronger.

Pet food attractant is a type of non-nutritive additive added to the primary pet feed to improve the palatability of pet feed, stimulate the appetite of the animal, and increase the feed intake [15]. As the consumption of prepackaged pet food increased, the requirement for attractants, which were specially designed to make pet feed, snacks, and supplements taste better, became more critical. However, there were few studies on pet palatability, and the production technology of pet attractants was relatively backward in China. In addition, the misuse of flavors and spices was also commonly used in attractants, which led to a relatively poor “tolerance” of pet feed in China.

Pet cats like to eat meat, so any item with meat flavor can be used as bait for pet cats. It has been reported that meat taste attractants could be divided into two categories: one where the substance had the meat aroma characteristic and the other which was prepared by the reaction. Koppel et al. [16] designed a pet food using fresh meat, including chicken fat and chicken by-product meal, as major ingredients and found out that fresh meat inclusion could effectively cover up the bitterness and increase its fishy flavor and cohesiveness. Parker et al. [17] added glucose and glycine to a pet food recipe (in the presence of cysteine) to study the change in pet food odor and found that glucose and glycine could increase the formation of 2-furfural and pyrazines, which were related to the toasty and nutty flavor. Moreover, due to the immature production technology of food attractants in China, the production cost of pet food attractants is high. Therefore, it is imperative to develop a complete set of technology for the production and processing of pet food attractants.

In this study, the cat food attractant was prepared by Maillard reaction using protein hydrolysate of grass carp scraps as raw material and glucose and cysteine hydrochloride as substrate. The scavenging rates of OH, O_2_^−^, and DPPH free radicals, as well as the ability of ferric ions, were determined to analyze the antioxidant activity of the self-made attractant. The volatile composition of cat food attractant was analyzed by GC-MS. Then, the application effect of self-made cat food attractant was investigated through the acceptance and palatability test of cats. The results showed that Maillard reaction could increase alcohols and aldehydes in self-made attractants and significantly increase antioxidant activity of the self-made attractants. Meanwhile, the acceptability and palatability of cat food were both significantly improved after adding the self-made attractants. These could provide the basis for further processing and high-value utilization of grass carp waste.

## 2. Materials and Methods

### 2.1. Sample

Grass carp waste was purchased from Nanjing Weigang food market and stored at 4 °C. Trypsin (4000 U/g), and flavor protease (200,000 U/g) were bought from Guangxi Nanning Pangbo Biological Co., Ltd. (Nanning, China). LY-MTG, a commercial cat food attractant, was supplied by Jiangsu Lianyi Biotechnology Co., Ltd. (Lianyungang, China). Basic cat food (unflavored) was obtained from ISCO Pet Food Group (Nanjing, China).

### 2.2. Chemicals

Glucose, cysteine hydrochloride, thiamine, sodium hydroxide, hydrochloric acid, formaldehyde, DPPH, trihydroxymethylaminomethane (Tris), pyrogarcinol, phenanthrene, phosphoric acid, hydrogen peroxide, phenanthrene, ferrous chloride, ascorbic acid, and other analytic reagents were procured from Sinopharm Chemical Reagent Co., Ltd. (Shanghai, China).

### 2.3. Preparation of Protein Hydrolysate from Grass Carp Waste

The preparation of protein hydrolysate from grass carp waste was according to our previous method with minor modifications [18]. Firstly, trypsin and flavor protease were mixed 3:1 to make a complex enzyme. Then, hydrolysate was prepared by enzymic hydrolysis of grass carp residues for 6 h; the hydrolysis condition was as follows: reaction temperature of 50 °C, pH 7.3, liquid-solid ratio of 4.2:1, and enzyme addition amount of 2.2%.

### 2.4. Preparation of Cat Food Attractor by Maillard Reaction

The reaction condition was shown in Table 1. The amount of 40 mL protein hydrolysate of grass carp waste was collected in a 100 mL conical flask, blended with 0.5 mL thiamine, 1 mL sodium dihydrogen phosphate, 4 mL glucose, and 1 mL cysteine hydrochloride, then pH was adjusted to 7.0 by adding 1 mol/L NaOH. After reacting in an oil bath at 115 °C, it was cooled and then stored in the refrigerator at 4 °C.

### 2.5. Antioxidant Activity Analysis

#### 2.5.1. DPPH Free Radical Scavenging Ability

The DPPH free radical scavenging ability of attractants was analyzed using the method of Saraswati et al. [19] with slight modification. The amount of 1.0 mL of different samples was mixed with 0.2 mL of 0.4 mmol/L DPPH solution (prepared with absolute ethanol). Then, 2.0 mL of deionized water was added and blended. After 30 min of reaction, the absorbance value at 517 nm was measured by UV spectrophotometer, as *A*_1_. In the blank group, the same amount of distilled water was used to replace the mixed solution, and *A*_5l7_ was measured as above, as *A*_0_. In the interference experiment, absolute ethanol was used to replace DPPH solution, and *A*_5l7_ was measured as *A*_2_ in the same treatment as above. The formula of DPPH free radical scavenging ability was as follows:(1)$$P(\%)=\left[A_{0}-(A_{1}-A_{2})\right]/A_{0}\times 100\%$$
where *P*% is the clearance rate.

#### 2.5.2. Hydroxyl Radical Scavenging Ability

According to the method described by Ajibola et al. [20], 1.5 mL phosphate buffer (pH = 7.4, 0.2 mol/L), 1.5 mL o-dinitrogen solution (1.0 mmol/L), and 1 mL FeSO_4_ solution (1.5 mmol/L) were blended with 2 mL distilled water in reaction tube. The reaction tube was kept in a 37 °C constant temperature water bath for 60 min, then the absorbance value was measured at 509 nm after cooling, as *A*_0_. The 2 mL distilled water in the above mixture was replaced by 1.0 mL hydrogen peroxide solution (0.01%) and 1.0 mL distilled water; *A*_509_ was read, marked as *A*_1_. Similarly, after mixing the same phosphate buffer, o-dinitrogen solution and FeSO_4_ solution with 1.0 mL hydrogen peroxide solution (0.01%) and 1.0 mL sample solution, *A*_509_ was recorded as *A*_2_.

Hydroxyl radical scavenging rate according to the formula:(2)$$P(\%)=(A_{2}-A_{1})/(A_{0}-A_{1})\times 100\%$$

#### 2.5.3. Fe^2+^ Chelation Ability

The analysis of Fe^2+^ chelation ability was performed by a previous method with some modifications [21]. The amount of 1.0 mL sample solution (1.0 mL distilled water was used as blank control) was mixed with 0.05 mL ferrous chloride (2.0 mmol/L), 0.2 mL ferrozine (5.0 mmol/L), and 2.75 mL distilled water for 10 min. UV spectrophotometer was used to detect the absorbance value at 562 nm, recorded as *A*_1_. The blank control was treated as *A*_0_ in the same way as above. Distilled water was used instead of ferrous chloride solution as an interference experiment; absorbance was measured at 562 nm, marked as *A*_2_. The chelation ability was calculated using the following formula:(3)$$P(\%)=\left[A_{0}-(A_{1}-A_{2})\right]/A_{0}\times 100\%$$

### 2.6. Volatile Compounds Analysis

The extraction of volatile compounds was carried out using the method operated by Luo et al. [22] with some changes. After thawing at 4 °C, the samples were rapidly cut into 1–2 mm pieces. Samples of 3.5 ± 0.2 g were put into a 40 mL headspace sample bottle and sealed with a silicone rubber mat; then, the bottle was maintained in a 40 °C constant temperature water bath for 30 min. The samples were then heated to 60 °C and purged with helium at 40 mL/min for 12 min. Tenax adsorbent was used for adsorption, the trap was heated to 220 °C, and the compounds were desorbed by helium at 220 °C for 2 min before going directly to gas chromatography. The trap should be kept at 240 °C for 30 min to remove possible residues or contaminants for the next injection.

J&W DB-5 quartz capillary column (60 m × 0.25 mm, 1μm) was used as chromatographic column, the flow rate of He as carrier gas was 1 mL/min, and temperatures of the inlet and interface were both 250 °C. The heating procedure was as follows: the temperature of GC oven started at 40 °C for 3 min, then increased to 130 °C at the rate of 5 °C/min, 8 °C/min to 200 °C, and lastly raised to 250 °C at 12 °C/min for 7 min; the split ratio was 1:10. Electron ionization source was used for mass spectrometry, ion source temperature was 280 °C, electron energy was 70 eV, excitation voltage was 350 V, emission current was 200 μA, and mass scan was in the range of 30–550 m/z. The mass spectrum was matched with MEANLIB, REPLIB, NISTDEMO, and Wiley Library for qualitative match retrieval, and the matching degree of more than 800 (maximum 1000) was used as basis for identification. The peak area normalization method was used to calculate the relative content of volatile compounds.

### 2.7. Evaluation of Application Effect of Cat Food Attractants

The results of the acceptance test of different groups of cat food (single-bowl feeding test) and the palatability test of different attractors (two-bowl feeding test) were measured to determine the application effect of Maillard reaction on self-made cat food attractors. The cats used in the experiment came from ISCO (Nanjing) Pet Food Group (30 cats).

Single-bowl feeding test: Thirty cats (15 Persian cats and 15 Oriental shorthair cats) were selected as experimental cats, and each experimental cat was provided with a food bowl. The amount of 200 g of cat food was added into the same group at 8 AM every day. The food bowl was removed at 3 PM the next day, the remaining cat food was weighed, and the feed intake of each cat was recorded. After 2 consecutive days of testing, the average value was taken to calculate the feeding rate of experimental cats with cat food.
Feeding rate (%) = feed intake/total cat food × 100%(4)

Two-bowl feeding test followed the previous method with a slight change [23]. Thirty cats (15 Persian cats and 15 Oriental shorthair cats) were selected as experimental cats, and 200 g of experimental and control cat food was added into two cleaned and sterilized food bowls, respectively. Bowls were placed in the cat carrier at 8 AM every day; the first food bowl that each cat touched and the first food bowl that was taken were recorded separately; the food bowl was removed at 3 PM the next day, then the remaining cat food was weighed; the experiment lasted for 2 days. The number of “first sniff” and “first bite” were counted, and the intake rates of different cat diets (%) were recorded.

### 2.8. Statistical Analysis

All data were statistically processed by Origin 8.0, and SPSS Version 19.0 (SPSS Inc., Chicago, IL, USA) was used for one-way analysis of variance (ANOVA); the least significant difference method was used to test the significance of difference between different means at a significance level of *p* < 0.05. All experiments were carried out at least 3 times and all the results were shown as the means ± standard deviation (n ≥ 3).

## 3. Results and Discussion

### 3.1. Effect of Maillard Reaction on Volatile Compounds

As shown in Figure 1, 64 volatile compounds were detected in the self-made attractants, which was lower than the 94 volatiles detected in the commercial attractants. The number of hydrocarbons, alcohols, ketoacids, esters, and other volatiles were 17, 21, 21, 16, and 8 in the commercial attractants, and 14, 15, 12, 4 and 6 in the self-made attractants, respectively. However, for aldehydes, 18 aldehydes were found in the self-made attractants, while only 11 were in the commercial attractants. Therefore, there was a great difference in the composition of volatile compounds between the self-made and the commercial attractants.

The relative content of different volatile components is presented in Figure 2, indicating that in the self-made attractants, alcohols and aldehydes were the main volatile components, accounting for 34.29% and 33.52% of the total, respectively, followed by ketoacids, representing 16.90%, while hydrocarbons, esters, and other classes accounted for 9.8%, 0.64%, and 4.84%, respectively. This result was similar to the dog food attractants studied by Chen et al. [15] who found 7 alcohols and 11 aldehydes in dog food attractants. In the commercial attractants, hydrocarbons and esters were the main volatiles, representing 23.30% and 28.58% of the total compounds, respectively, followed by alcohols and aldehydes, accounting for 17.87% and 15.19%, respectively; keto acids and other classes were the lowest volatile compounds, which made up 10.57% and 4.49% of the total, respectively. This could be due to the difference in raw materials and the process of Maillard reaction, such as animal fat [16], grains [24], antioxidants [25], and other oxidative reactions during the process. Moreover, the addition of flavoring ingredients in the commercial attractants could also be the reason for the difference in flavor between them.

#### 3.1.1. Hydrocarbon

The hydrocarbons were generally composed of alkanes and alkenes. The alkanes had little contribution to food flavor due to their relatively high threshold level; however, alkenes could provide a fresh smell and fruit flavor to the product due to the low threshold of them [26]. In addition, alkenes were easily decomposed by oxidation, resulting in small molecular hydrocarbons, alcohols, and carbonyl aroma components [27]. As shown in Table 2, the composition of hydrocarbons in two cat food attractants was quite different; only tetradecane, nonadecane, and 1-limonene were the same compounds in both groups. Among them, 1-limonene with a special lemon flavor had a low threshold which could bring a sense of pleasure [26]. As for the composition of other hydrocarbons detected in two attractants, short-chain alkenes such as 1, 3-butadiene, 2, 4-heptadiene, (Z) 2-octene, (E)-2-decene contributed more to the flavor in self-made cat food attractants; nevertheless, benzene ring, 2-ethyl-1-decene,2, and 2-dimethyl-4-decene were the main hydrocarbons in commercial attractants.

#### 3.1.2. Alcohol

Alcohol played an important role in the flavor composition of pet food attractants which could be generated from Maillard reactions and lipid oxidation [28]. Bermudez et al. [29] reported that alcohols could give products an elegant aroma; meanwhile, these substances were suitable solvents for other aroma substances and could be chemically changed into different components. Due to their low olfactory threshold and high aroma value, they played a crucial part in the formation of the overall aroma. As can be seen from Table 3, there were 10 types of the same alcohols in both of two types of attractants, and all of them were of low molecular weight alcohols, which contributed significantly to the flavor. The alcohols in the two attractants were mostly unsaturated alcohols, such as 1-octene-3-ol and 2-octenol, with a low odor threshold, presenting a pleasant mushroom flavor [30]. In addition, furfuryl alcohol detected in the commercial attractant was a type of furan alcohol with a meat aroma, which was a good organic solvent and could significantly improve the flavor quality of the product [31].

#### 3.1.3. Aldehyde

The threshold value of aldehydes is low, and they come mainly from the oxidation of lipids as well as Maillard reactions and Strecker degradation, contributing to meaty, floral, and fruity flavor. Still, the stability of aldehydes in food system is poor, and most of them are the intermediates of volatile substances [22]. As can be seen from Table 4, there were six identical aldehydes detected in two attractants, namely hexal, heptanal, octanal, nonal, benzaldehyde, and 2-octenal, which are responsible for beef fat and fat fragrance flavor. Other studies have shown that these short-chain aldehydes had an important contribution to the formation of meat flavor and were the key components of the flavor of cooked meat products [27,32]. Moreover, the mixture of hexal, benzaldehyde, and other medium-chain aldehydes could significantly promote the flavor of fish meat [33]. Furfural is a derivative of furan substances, with barbecue flavor of typical aldehydes, and its threshold is shallow [15]. It could obviously enhance the odor of the product and produce keto acid derivatives quickly, which was detected only in self-made attractants.

#### 3.1.4. Keto Acids

Acids and ketones were the final products of the oxidation of hydrocarbons and were typical volatile compounds in Maillard reaction products [34]. In particular, the unsaturated acids and ketones contributed to the flavor, mainly presenting fat and cream aroma [30]. Table 5 shows the analysis of acids and ketones in different attractants; five acids and ketones, namely 2-octanone, 1-octene-3-ketone, n-octanoic acid, nonanoic acid, and heptanoic acid, were found to be identical in two kinds of attractants. Among them, 2-ketones, including 2-octanone, have noticeable blue cheese and fat flavor, which could be derived from the oxidation of free fatty acids and Maillard reactions [30]. Acid components such as nonanoic acid and heptanoic acid have a sour taste and are the common flavor ingredients in cat feed attractants. In addition, n-octanoic acid, nonanoic acid, and heptanoic acid were always found in meat products, which came from the Maillard reaction and hydrolysis of lipids during processing [26].

#### 3.1.5. Esters and Others

Esters could significantly bring fruity and fat aroma and enhance the flavor quality of products [27]; however, the presence of esters in self-made attractants was limited. Only 4 esters were observed in self-made attractant while, on the contrary, 16 esters were detected in commercial attractant. In terms of the total relative content of esters, the self-made attractants were only 0.638%, but esters were the most abundant flavor ingredients in commercial attractants. This might be the difference in Maillard reaction process and raw material substrate; for example, yeast extract and yeast powder were used in commercial attractants [24,35]. Other possible reasons might be due to the shorter ripening time of the self-made attractants and the flavoring ingredients added into the commercial attractant to enhance the sensory characteristics.

In addition to the above volatile compounds, the other volatiles monitored in the two attractors were mainly furans and pyrazines, and the distinction of these components is primarily caused by the different processes of Maillard reaction. Generally, furan and its derivatives have bean, fruit, fragrance, and vegetable odor was often found during the ripening period in most meat products [36]. As shown in Table 6, pyrazines were not detected in self-made attractors, and there were only two of the same furan compounds existed in the two attractants.

### 3.2. Antioxidant Activity Analysis of Food Attractants

#### 3.2.1. DPPH Free Radical Scavenging Ability

DPPH free radical was a stable one-electron radical with a strong absorption peak at 517 nm, which gradually disappeared when the scavenger was paired with a single electron from DPPH radical. The DPPH free radical scavenging ability of samples in different groups is presented in Figure 3. The scavenging abilities of groups 1 to 4 against DPPH free radicals were 21.38%, 13.22%, 43.41%, and 56.73%, respectively. There was no significant difference between group 1 and 2 (*p* > 0.05), but they were both significantly lower than group 3 (*p* < 0.05), which indicated that adding glucose and cysteine hydrochloride could significantly (*p* < 0.05) increase the scavenging ability of protein hydrolysate from fish scraps (self-made group) on DPPH free radical through Maillard reaction. Similarly, Affes et al. [12] reported that after adding glucose, the heated chitosan films had greater DPPH radical scavenging activity than unheated films without glucose. The reason could be due to the MRPs formed in self-made cat food bait, which enhanced the ability of the attractor to donate hydrogen atoms, resulting in stabilizing the free radicals [37]. However, the scavenging capacity of self-made attractors (group 3) on DPPH free radical was significantly lower than that of commercial cat food attractors (group 4). In addition, 0.2 mg/mL Vitamin C (ascorbic acid (AA)) solution (group 5) (*p* < 0.05), which could be attributed to the addition of antioxidants such as TBHQ, rosemary, and α-tocopherol in commercial attractants, led to the extension of preservation time. AA itself, with a strong antioxidant capacity, has a significant effect on scavenging free radicals [38]. Hence, their total antioxidant activity was higher than that of our self-made attractants.

#### 3.2.2. Hydroxyl Radical Scavenging Ability

Hydroxyl radical (OH) scavenging rate is a significant indicator of antioxidant capacity. As shown in Figure 4, the scavenging abilities of OH in groups 1 to 4 were 5.44%, 3.89%, 15.12%, and 21.78%, respectively. Similar to the results of DPPH radical scavenging ability, there was no significant difference between group 1 and group 2 (*p* > 0.05), but both were significantly lower than that of group 3 (*p* < 0.05), indicating that Maillard reaction could significantly (*p* < 0.05) increase the antioxidant activity of fish waste protein hydrolysate (self-made group). Peng et al. [39] noted that the scavenging ability of ·OH was related to the Fe^2+^-chelation activity, since ·OH was often produced through iron and hydrogen peroxide. Similar to the above, the hydroxyl radical scavenging ability of self-made attractants (group 3) was significantly lower than that of commercial attractants (group 4) and AA solution (group 5) (*p* < 0.05).

#### 3.2.3. Fe^2+^ Chelation Ability

Since the chelating ability of scavengers to Fe^2+^ was proportional to the scavenging ability of OH radical [40], the Fe^2+^ chelation ability was subsequently studied further (Figure 5). The Fe^2+^ chelation ability of different groups was in agreement with the scavenging ability of ·OH radical, manifested as Fe^2+^ chelation. The ability of group 3 was significantly higher than both groups 1 and 2 (*p* < 0.05) yet significantly (*p* < 0.05) lower than that of commercial cat food attractor (group 4) and 0.2 mg/mL EDTA solution.

Therefore, these results indicate that glucose addition could significantly increase the antioxidant activity of fish waste protein hydrolysate. Furthermore, a remarkably promotion was found after Maillard reaction processing. The main reason was that after Maillard reaction, there was a large number of furans and their derivatives, ketones, and acids in the products, so MRPs had a certain free radical scavenging ability.

In the commercialization of our final product, we would add various natural antioxidants such as turmeric microcapsules, allicin microcapsules, active polysaccharides, etc., to enhance the antioxidant activity of our self-made pet food attractant.

### 3.3. Application Effect of Self-made Cat Food Attractant

Unlike the sensory evaluation of human food, it is difficult for pets to express their preferences to us. Hence, we used some techniques to detect pets’ behavior at mealtime to see whether they liked the food. The measure of food intake was palatability, which reflected the degree of preference and reception of food. Two of the most-used tests are single-bowl and two-bowl assays [41].

#### 3.3.1. Analysis of Acceptability of Cat Food Attractant

Single-bowl feeding assay was used to analyze the cat’s acceptance of the sample by grazing rate, as pet cats could only eat one type of food at a meal without choice. Thus, this assay could reflect the influence of food attractants on the smell, taste, or texture of blank cat food. Compared with the control group (blank cat food), the intake rate of group A significantly (*p* < 0.05) increased to 75.37% (Table 7), indicating that the self-made attractant could remarkably (*p* < 0.05) increase the acceptance of cat food. Additionally, there was no significant difference in feeding rate between group A and group B (*p* > 0.05), which meant there was no significant difference in acceptability between self-made and commercial cat food attractors. Acceptability is the ability to maintain weight and performance through food intake to gain enough energy, regardless of taste or aroma of the food; here, the influence of different volatiles composition on acceptability between two attractants was not evident.

#### 3.3.2. Palatability Analysis of Cat Food Attractant

Another experiment for palatability was two-bowl feeding, which was used mainly to evaluate the preference or degree of liking of two kinds of food. It could be analyzed from three aspects: first sniffing, first bite, and feeding rate, which was closely related to the aroma of food [42]. Table 8 showed the results of the palatability test between self-made and commercial cat food attractors. During two consecutive days of feeding experiments, the first sniffing and first bite in group A were lower than those in group B. According to the analysis of volatile compounds, the possible reason was that the number of volatile substances of the self-made attractant were much less than those of the commercial attractant, which resulted in insufficient quantity in the first sniffing and first bite. Since cats prefer more acidic foods, more abundant ketoacids could have a positive effect on palatability [42]. Furthermore, there was no significant difference in feeding rate between the first day and the second day in either group A or B, indicating that the operation error of the two-bowl feeding experiment was small. The feeding rate of group A was significantly lower than that of group B (*p* < 0.05), which might be due to the simple composition of the substrate used in self-made attractants and the meat-based flavors, yeast, and various vitamins added in commercial attractants. In the subsequent experiment, animal and plant protein and yeast extract could be added to promote the odor, to increase the bait effect on pet cats.

## 4. Conclusions

In this study, the cat food attractant was prepared through Maillard reaction with glucose and cysteine hydrochloric acid as substrate, using protein enzymatic hydrolysate of grass carp waste as raw material. It was demonstrated that both self-made and commercial attractants had obvious characteristic volatiles, but there were significant differences in volatile compounds composition between them. Alcohols and aldehydes were the highest relative percentage volatiles in self-made attractants, while hydrocarbons and esters were the most volatile substances in commercial attractants. The results showed that MRPs in this study could significantly increase the scavenging ability of OH and DPPH radical and the chelating ability of Fe^2+^. According to the acceptance and preference test, both self-made and commercial attractants could improve the feeding rate of blank cat food, and there was no significant difference in acceptability between commercial and self-made attractants. However, the self-made attractant had significantly lower indexes of first smell, first bite, and feeding rate than the commercial group. These results indicated that self-made attractants could effectively enhance the feeding rate of blank cat food, but the effect was lower than that of commercial attractants. This might be due to the difference in volatile substances and the Maillard reaction processing with natural raw materials in this study, without adding other umami ingredients, flavors, or spices.

## Figures and Tables

**Figure 1 molecules-27-07239-f001:**
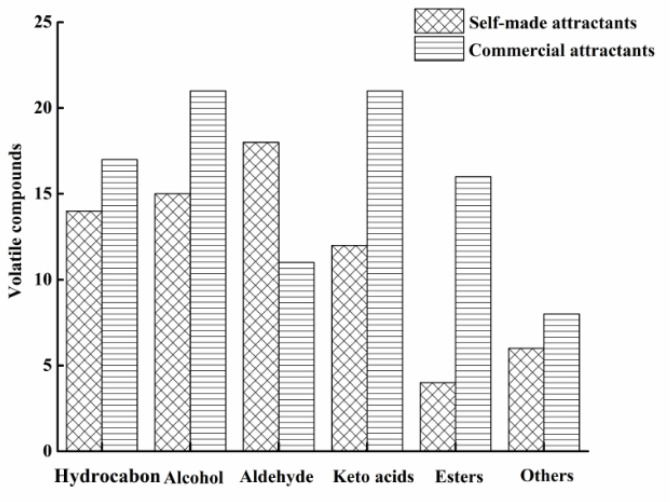
Composition of volatile compounds in self-made and commercial attractants.

**Figure 2 molecules-27-07239-f002:**
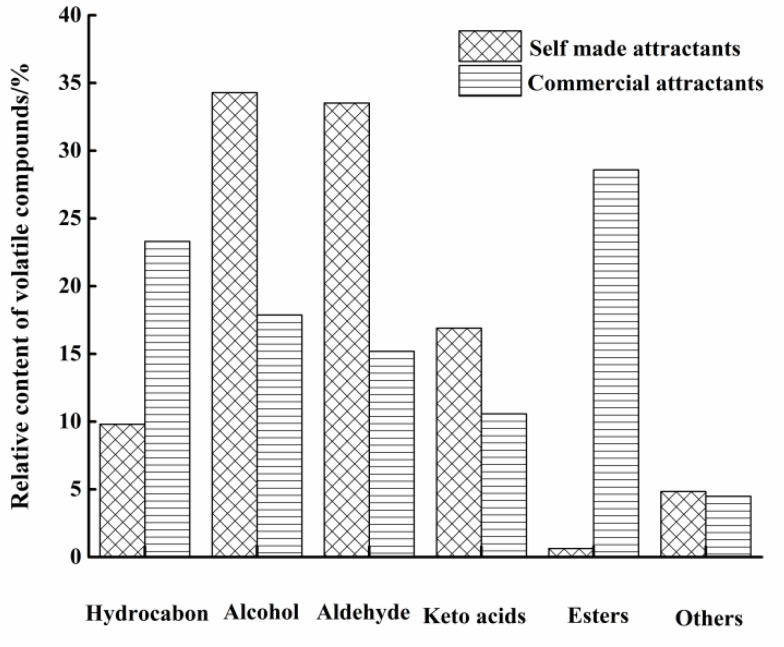
Relative content of volatile compounds in self-made and commercial attractants.

**Figure 3 molecules-27-07239-f003:**
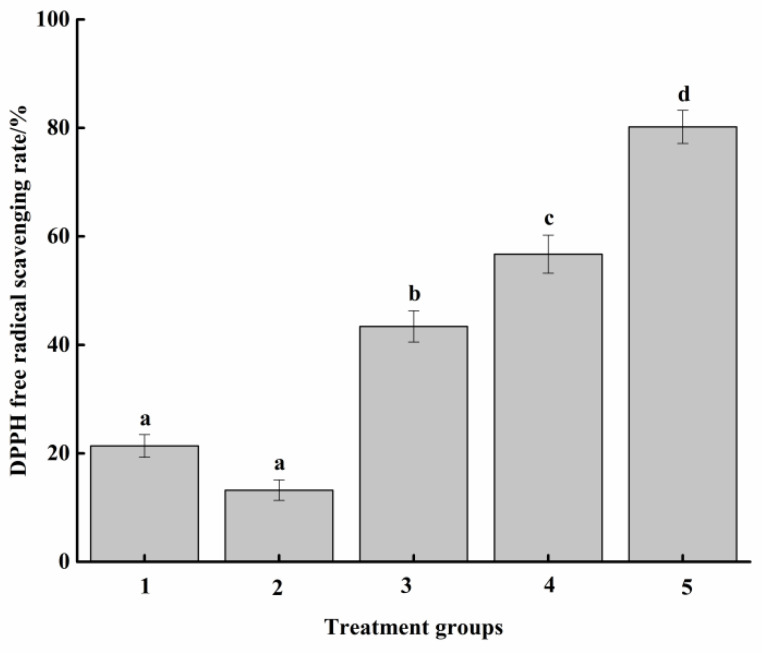
Scavenging ability of DPPH free radical. Group 1: Protein hydrolysate of grass carp waste. Group 2: Enzymatic hydrolysis solution of grass carp waste (without adding cysteine hydrochloride and glucose). Group 3: Self-made cat food attractant produced by Maillard reaction. Group 4: Commercial cat food attractant. Group 5: AA solution (0.2 mg/mL). Different letters between groups represent significant differences (*p* < 0.05).

**Figure 4 molecules-27-07239-f004:**
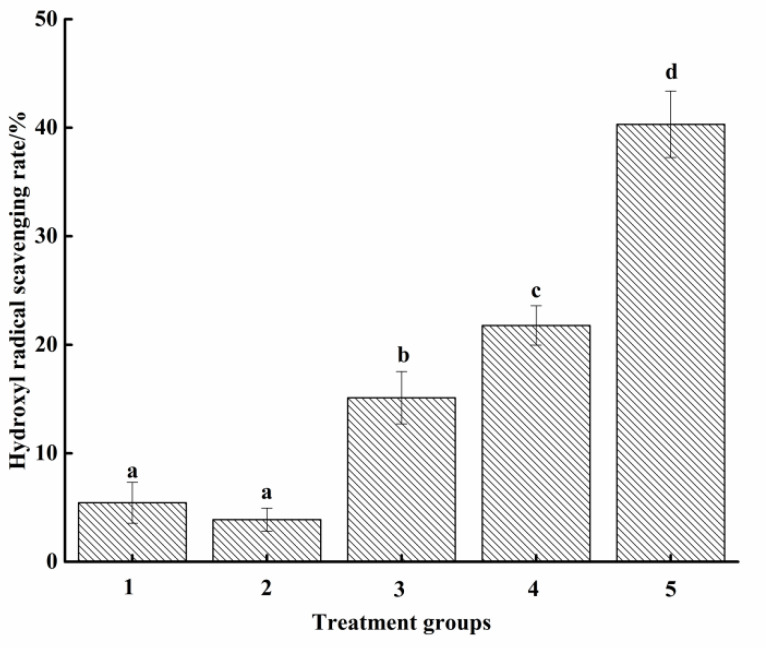
Scavenging ability of hydroxyl radical. Groups 1 to 5 were the same as above. Different letters between groups represent significant differences (*p* < 0.05).

**Figure 5 molecules-27-07239-f005:**
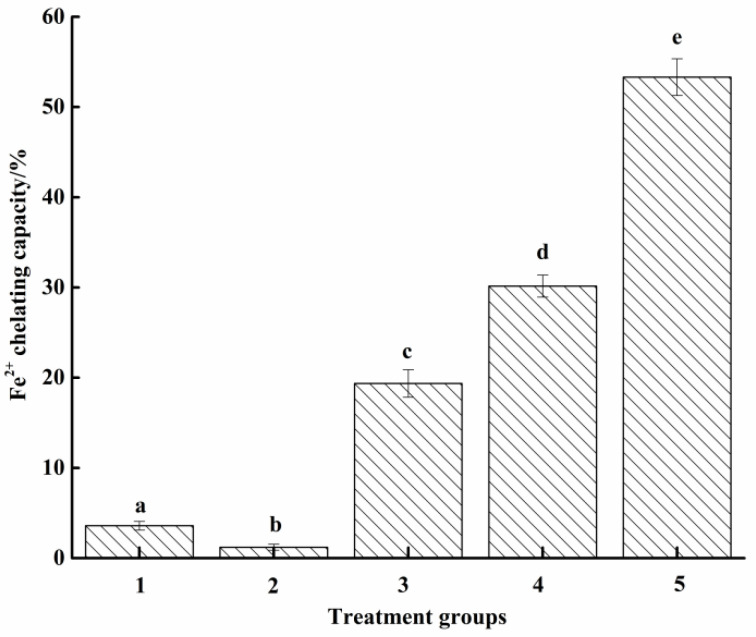
Chelating ability of Fe^2+^. Groups 1 to 4 were the same as above. Group 5: EDTA solution (0.2 mg/mL). Different letters between groups represent significant differences (*p* < 0.05).

**Table 1 molecules-27-07239-t001:** Maillard reaction parameters of self-made attractants.

Factors	Reducing Sugar (%)	Cysteine Hydrochloride (%)	Thiamine (%)	Na_2_HPO_4_ (%)	Temperature (°C)	Time (min)	pH
Parameter	4	1	0.5	1	115	45	7.0

**Table 2 molecules-27-07239-t002:** Hydrocarbons in self-made and commercial attractants (percentage of the total peak area).

Hydrocarbons	Self-Made Attractants		Commercial Attractants	
	Volatile Compounds	Relative Content (%)	Volatile Compounds	Relative Content (%)
1	Tetradecane	0.041 ± 0.01 ^e^	Tetradecane	2.199 ± 0.31 ^b^
2	Nonadecane	0.902 ± 0.02 ^b^	Nonadecane	0.991 ± 0.04 ^c^
3	1-Limonene	0.397 ± 0.01 ^c^	1-Limonene	2.281 ± 0.32 ^b^
4	Octane	0.870 ± 0.02 ^b^	2,2,4,6,6-Pentamethylheptane	8.456 ± 0.45 ^a^
5	Trichloromethane	0.342 ± 0.00 ^c^	Decane	1.999 ± 0.21 ^b^
6	Ethylene oxide	0.436 ± 0.03 ^c^	2,2,4,4-Tetramethyloctane	1.922 ± 0.26 ^b^
7	Hexadecane	0.424 ± 0.05 ^c^	6-Methyl-tridecane	0.799 ± 0.05 ^c^
8	Octodecane	0.132 ± 0.02 ^d^	2,5,9-Trimethyl-decane	0.187 ± 0.03 ^e^
9	3-(propyl-2-enoyloxy) dodecane	0.104 ± 0.05 ^d^	Undecane	2.181 ± 0.28 ^b^
10	1,3-Butadiene	0.177 ± 0.01 ^a^	Dodecyl-cyclopropyl siloxane	0.114 ± 0.17 ^e^
11	2,4-heptadiene	1.301 ± 0.07 ^b^	2,3-Dihydroxybutane	0.682 ± 0.02 ^c^
12	(*Z*) 2-Octene	0.456 ± 0.03 ^c^	3-Methyl-tridecane	0.312 ± 0.02 ^d^
13	1,6,10-Hexadecatene	0.653 ± 0.05 ^c^	Ethylbenzene	0.259 ± 0.01 ^d e^
14	(*E*)-2-decene	3.574 ± 0.75 ^a^	Xylene	0.083 ± 0.00 ^e^
15			2-Ethyl-1-decene	0.301 ± 0.01 ^d^
16			2,2-Dimethyl-4-decene	0.124 ± 0.01 ^e^
17			1-Tetradecene	0.408 ± 0.05 ^d^
Total		9.812 ± 1.01		23.299 ± 6.33

Data are means ± SEM (n = 4). Data with different letters in the same row are significantly different (*p* < 0.05).

**Table 3 molecules-27-07239-t003:** Alcohols in self-made and commercial attractants (percentage of the total peak area).

Alcohols	Self-Made Attractants		Commercial Attractants	
	Volatile Compounds	Relative Content (%)	Volatile Compounds	Relative Content (%)
1	1-Pentene-3 alcohol	0.124 ± 0.02 ^e^	1-Pentene-3 alcohol	0.138 ± 0.02 ^d^
2	* N * -pentanol	2.673 ± 0.15 ^b^	* N * -pentanol	0.830 ± 0.07 ^b^
3	1,5-Octandiene-3-ol	0.855 ± 0.09 ^c^	1,5-Octandiene-3-ol	0.295 ± 0.03 ^d^
4	1-Octanol	4.679 ± 0.34 ^b^	1-Octanol	0.316 ± 0.05 ^c d^
5	2,4-Dimethyl-cyclohexanol	0.403 ± 0.05 ^d^	2,4-Dimethyl-cyclohexanol	4.057 ± 0.56 ^a^
6	* N * -heptanol	2.845 ± 0.30 ^b^	* N * -heptanol	0.065 ± 0.00 ^e^
7	Hexyl alcohol	3.726 ± 0.46 ^b^	Hexyl alcohol	0.529 ± 0.06 ^c^
8	1-Octene-3-ol	9.360 ± 0.81 ^a^	1-Octene-3-ol	3.133 ± 0.66 ^a^
9	2-Octene alcohol	2.633 ± 0.17 ^b^	2-Octene alcohol	0.624 ± 0.05 ^b c^
10	Benzyl alcohol	0.479 ± 0.05 ^d^	Benzyl alcohol	0.061 ± 0.00 ^e^
11	1-Tridecane-1-ol	0.550 ± 0.06 ^d^	1-Heptene-1-ol	0.232 ± 0.01 ^d^
12	3-Heptene-1-ol	2.947 ± 0.25 ^b^	Heptanol	0.453 ± 0.05 ^c^
13	2,7-Octandiene-1-ol	0.328 ± 0.05 ^d^	2-Hexadecanol	0.613 ± 0.07 ^b c^
14	3-Nonene-1-ol	1.102 ± 0.08 ^c^	2-Tetradecanol	0.161 ± 0.02 ^d^
15	4-Ethyl cyclohexanol	1.587 ± 0.65 ^c^	* N * -caprylic alcohol	0.871 ± 0.08 ^b^
16			(+)-5-Methyl-2-hexanol	0.426 ± 0.05 ^c^
17			2-Cyclopropyl 1-propanol	0.208 ± 0.03 ^d^
18			Furfuryl alcohol	1.302 ± 0.35 ^b^
19			2,4-Dimethyl-cyclohexanol	0.138 ± 0.02 ^d^
20			2-Phenethyl alcohol	3.269 ± 0.29 ^a^
21			3,7-Dimethyl-1-octanol	0.149 ± 0.02 ^d^
Total		34.292 ± 3.05		17.868 ± 2.07

Data are means ± SEM (n = 4). Data with different letters in the same row are significantly different (*p* < 0.05).

**Table 4 molecules-27-07239-t004:** Aldehydes in self-made and commercial attractants (percentage of the total peak area).

Aldehydes	Self-Made Attractants		Commercial Attractants	
	Volatile Compounds	Relative Content (%)	Volatile Compounds	Relative Content (%)
1	Hexanal	3.758 ± 0.66 ^b^	Hexanal	3.912 ± 0.38 ^a^
2	Heptanal	1.222 ± 0.31 ^c^	Heptanal	0.544 ± 0.04 ^c^
3	Octanal	2.373 ± 0.25 ^b,c^	Octanal	0.449 ± 0.05 ^c^
4	Nonanal	6.699 ± 0.75 ^a^	Nonanal	0.971 ± 0.09 ^b^
5	2-Octene aldehyde	0.144 ± 0.02 ^e^	2-Octene aldehyde	0.524 ± 0.05 ^c^
6	Benzaldehyde	0.145 ± 0.03 ^e^	Benzaldehyde	1.497 ± 0.20 ^b^
7	* E * -2-heptene aldehyde	4.687 ± 0.55 ^a,b^	2-Methylpropyl aldehyde	1.124 ± 0.18 ^b^
8	2,4-Heptanedienal	0.296 ± 0.03 ^d^	Isovaleraldehyde	5.721 ± 0.65 ^a^
9	2-Octene aldehyde	4.015 ± 0.46 ^a^	2-Pentyl-2-nonenal	0.119 ± 0.25 ^d^
10	Decanal	0.258 ± 0.02 ^d^	α-Ethylene-phenylacetaldehyde	0.102 ± 0.02 ^d^
11	Benzaldehyde	1.181 ± 0.20 ^c^	5-Methyl-2-(1-methylethyl)-2-hexenal	0.226 ± 0.09 ^d^
12	(*Z*)-6-Nonene aldehyde	2.312 ± 0.25 ^b,c^		
13	Dimethyl-silane dialdehyde	0.692 ± 0.07 ^d^		
14	4-Ethyl benzaldehyde	0.252 ± 0.03 ^a^		
15	Trans-undecane-2-enal	1.313 ± 0.05 ^c^		
16	(*E*,*E*)-2,4-decanodienal	0.963 ± 0.08 ^c^		
17	2,4-Decanodienal	2.845 ± 0.36 ^b,c^		
18	Furfuraldehyde	0.362 ± 0.05 ^d^		
Total		33.515 ± 3.25		15.189 ± 1.85

Data are means ± SEM (n = 4). Data with different letters in the same row are significantly different (*p* < 0.05).

**Table 5 molecules-27-07239-t005:** Keto acids in self-made and commercial attractants (percentage of the total peak area).

Keto Acids	Self-Made Attractants		Commercial Attractants	
	Volatile Compounds	Relative Content (%)	Volatile Compounds	Relative Content (%)
1	2-Octanone	0.168 ± 0.02 ^e^	2-Octanone	0.160 ± 0.02 ^c^
2	1-Octene-3-ketone	2.620 ± 0.31 ^b,c^	1-Octene-3-ketone	0.517 ± 0.05 ^b^
3	* N * -caprylic acid	6.834 ± 0.75 ^a^	* N * -caprylic acid	2.667 ± 0.31 ^a^
4	Nonanoic acid	0.198 ± 0.25 ^e^	Nonanoic acid	0.082 ± 0.00 ^d^
5	Heptylic acid	1.690 ± 0.15 ^c^	Heptylic acid	0.169 ± 0.02 ^c^
6	3-Hexene-2-ketone	0.655 ± 0.77 ^d^	2,3-Pentarone	0.251 ± 0.01 ^c^
7	2-Heptanone	0.589 ± 0.60 ^d^	2-Hydroxy-acetone	0.387 ± 0.03 ^b,c^
8	3-Octene-2-ketone	0.283 ± 0.03 ^e^	Hydroxy acetone	0.679 ± 0.00 ^b^
9	3-Nonene-2 ketone	0.240 ± 0.02 ^e^	2,2,6-Trimethyl-cyclohexanone	0.242 ± 0.02 ^c^
10	Linolenic acid	0.369 ± 0.05 ^e^	2,3-Diketone	1.603 ± 0.08 ^a^
11	Hexanoic acid	1.841 ± 0.21 ^c^	6-Methyl-5-heptene-2-ketone	0.498 ± 0.05 ^b^
12	Acetic acid	1.413 ± 0.17 ^c^	2-Nonyl ketone	0.098 ± 0.01 ^c,d^
13			3,6-Dimethyl-oct-2-ketone	0.050 ± 0.00 ^d^
14			Geranylacetone	0.041 ± 0.00 ^d^
15			2-Pyrrolidone	0.063 ± 0.01 ^d^
16			2-Methyl-2-ethyl-1-propyl propionic acid	0.329 ± 0.05 ^b,c^
17			2-Isobutyric acid	0.192 ± 0.02 ^c^
18			N-decanoic acid	0.981 ± 0.05 ^a^
19			9-Decanoic acid	0.103 ± 0.02 ^c^
20			Pentanoic acid	1.011 ± 0.04 ^a^
21			4-Methylvaleric acid	0.450 ± 0.05 ^b^
Total		16.900 ± 1.33		10.574 ± 1.45

Data are means ± SEM (n = 4). Data with different letters in the same row are significantly different (*p* < 0.05).

**Table 6 molecules-27-07239-t006:** Esters and other substances in self-made and commercial attractants (percentage of the total peak area).

Esters	Self-Made Attractants		Commercial Attractants	
	Volatile Compounds	Relative Content (%)	Volatile Compounds	Relative Content (%)
1	* N * -pentylbutyllactone	0.071 ± 0.01 ^b^	* N * -pentylbutyllactone	0.069 ± 0.00 ^f^
2	Methyl 2-hydroxyisobutyrate	0.052 ± 0.01 ^b^	Propyl 2-hydroxypropionate	0.214 ± 0.03 ^e^
3	* P * -caproic acid-p-nitrophenyl ester	0.266 ± 0.03 ^a^	Butyl acetate	0.306 ± 0.05 ^e^
4	2-Hexene-4-lactone	0.249 ± 0.03 ^a^	Buty propionate	0.265 ± 0.02 ^e^
5			Decyl-butyl phthalate	0.637 ± 0.05 ^e^
6			Butyl acrylate	1.652 ± 0.20 ^d^
7			Butyl butyrate	1.127 ± 0.15 ^d^
8			Ethyl caprylate	6.710 ± 0.55 ^b^
9			Ethyl caprate	11.037 ± 0.97 ^a^
10			Ethyl 9-decenoate	1.126 ± 0.16 ^d^
11			Octanoic acid-3-Methyl butyl ester	0.419 ± 0.45 ^a^
12			Ethyl laurate	3.265 ± 0.35 ^c^
13			3-Methylbutyrate	0.718 ± 0.07 ^d^
14			* E * -11-hexadecanoenoic acid ethyl ester	0.875 ± 0.08 ^d^
15			Ethyl palmitate	0.166 ± 0.02 ^e^
Total		0.638 ± 0.05		28.585 ± 2.11
**Others**	**Self-Made Attractants**		**Commercial Attractants**	
	**Volatile Compounds**	**Relative** **Content (%)**	**Volatile Compounds**	**Relative** **Content (%)**
1	2-*N*-pentylfuran	2.459 ± 0.25 ^a^	2-*N*-pentylfuran	2.045 ± 0.21 ^a^
2	2-Acetylfuran	1.619 ± 0.17 ^a^	2-Acetylfuran	0.633 ± 0.05 ^b^
3	2-Heptyl furan	0.246 ± 0.02 ^b^	2-Acetyl pyrrole	0.321 ± 0.03 ^b^
4	2-Ethyl furan	0.114 ± 0.15 ^b^	2-Methylpyrazine	0.635 ± 0.06 ^b^
5	4-Methyl-5-hydroxyethyl thiazole	0.270 ± 0.02 ^b^	2, 6-Dimethylpiperazine	0.276 ± 0.02 ^b^
6	5-Pentyl-2-(5*H*) furan	0.136 ± 0.02 ^b^	2-Ethyl-6-methyl-pyrazine	0.166 ± 0.01 ^b^
7			2-ethyl-5-methyl-tetrahydrofuran	0.113 ± 0.01 ^b^
8			Nucleoside chrysanthemum ring	0.299 ± 0.04 ^b^
Total		4.843 ± 0.35		4.487 ± 0.38

Data are means ± SEM (n = 4). Data with different letters in the same row are significantly different (*p* < 0.05).

**Table 7 molecules-27-07239-t007:** Acceptability of cat food attractants in different groups. Group A: Blank cat food + 3% self-made attractants. Group B: Blank cat food + 3% commercial attractants. Group CK: Blank cat food. Different letters in different groups indicate significant differences (*p* < 0.05).

Treatment Groups	A	B	CK
Feeding rate/%	75.37 ± 7.32 ^a^	80.07 ± 8.33 ^a^	37.63 ± 10.37 ^b^

Data are means ± SEM (n = 4). Data with different letters in the same row are significantly different (*p* < 0.05).

**Table 8 molecules-27-07239-t008:** Preference of cat food attractants in different groups. Groups A, B, and CK were the same as above.

Treatment Groups	First Sniffing			First Bite			Feeding Rate/%		
	A	B	CK	A	B	CK	A	B	CK
First day	5	25	0	3	27	0	27.85 ± 9.31 ^b^	69.52 ± 7.49 ^a^	0.00 ± 0.00 ^c^
Second day	6	24	0	4	26	0	22.68 ± 6.77 ^b^	75.51 ± 8.52 ^a^	0.00 ± 0.00 ^c^

Data are means ± SEM (n = 4). Different letters indicate significant difference in feeding rate (*p* < 0.05).

## Data Availability

Not applicable.

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
