# Peer review of "Effects of Maillard Reaction on Volatile Compounds and Antioxidant Capacity of Cat Food Attractant"

_molecules, 2022, doi:10.3390/molecules27217239_

Round 1

Reviewer 1 Report

The study gives a thorough protocol for production of of a pet food attractant made from fish by-products by use of a Maillard reaction. Not many similar studies have been published to my knowledge. Interesting results concerning antioxidant ability. The results consist of detailed table presentations of chemical components obtained from the processing. Table content could reduced, but that must be up to the editor to decide. Figures are not self-explanatory, please check that. 

Check heading for Table 7. Two groups have the same notification. Which is commercial and which is home-made? Goes for the rest of the tables. 

Author Response

RESPONSE TO REVIEWER’S COMMENTS

Ref. No.: molecules-1965662

Title: Effects of Maillard reaction on volatile compounds and antioxidant capacity of cat food attractant

  Thank you for your comments concerning our manuscript (molecules-1965662), and those comments are all valuable and very helpful for revising and improving our paper. We have studied the comments carefully and have made corrections which we hope that will meet with approval. The changes in the revised manuscript have been marked in “Colored Text”. Here are the detailed responses addressing point to point according to the order of the comments.

Reviewer: 1 

  1. The study gives a thorough protocol for production of of a pet food attractant made from fish by-products by use of a Maillard reaction. Not many similar studies have been published to my knowledge. Interesting results concerning antioxidant ability. The results consist of detailed table presentations of chemical components obtained from the processing. Table content could reduced, but that must be up to the editor to decide. Figures are not self-explanatory, please check that.

Check heading for Table 7. Two groups have the same notification. Which is commercial and which is home-made? Goes for the rest of the tables.

Response: Thanks for the suggestion. This is an error in original manuscript caused by our carelessness. Group A: Blank cat food + 3% self-made attractants. Group B: Blank cat food + 3% commercial attractants. Group CK: Blank cat food. And we have checked the rest of the tables, including table 8.

We really appreciate the Editor and Reviewers’ work, and we hope the revised manuscript will meet the requirements of publication.

Yours sincerely,

Ji Luo, Ph. D

College of Life Science, Anhui Normal University.

Reviewer 2 Report

In this study, self-made cat-food attractant was prepared by Maillard reaction using protein hydrolysate of grass carp scraps as raw material. This research has certain practical significance and application value, but there are some problems. The article mainly described the phenomenon and experimental result, there was not much related discussion of the mechanism, the depth of the article was not enough.

1.     The introduction is not concise enough and it should be refined.

2.     As compared with the commercial cat-food attractant, the self-made cat-food attractant has weaknesses in first smell, first bite, and feeding rate, so I think the formulation of cat-food attractants needs to be further improved.

3.     The qualitative and quantitative methods of volatile substances are not accurate, only spectrum library was used for qualitative and peak area normalization method was used for semi-quantification in this study.

4.     The composition and content of volatile substances were only analyzed in this study. However, the contribution of volatile substances depends not only on their composition and content, but also in combination with other methods such as sniffing to identify the key volatile substances.

5.     In Table 5, the format of the content needs to be modified.

6.     P341, change “Scheme 3.” to “Group 3:”. Please be consistent.

7.     In Table 7, groups A and B both stand for the commercial attractants? Also, in table 8, what do groups A and B represent for?

8.     Please keep the reference format consistent.

Author Response

RESPONSE TO REVIEWER’S COMMENTS

Ref. No.: molecules-1965662

Title: Effects of Maillard reaction on volatile compounds and antioxidant capacity of cat food attractant

  Thank you for your comments concerning our manuscript (molecules-1965662), and those comments are all valuable and very helpful for revising and improving our paper. We have studied the comments carefully and have made corrections which we hope that will meet with approval. The changes in the revised manuscript have been marked in “Colored Text”. Here are the detailed responses addressing point to point according to the order of the comments.

Reviewer: 2

In this study, self-made cat-food attractant was prepared by Maillard reaction using protein hydrolysate of grass carp scraps as raw material. This research has certain practical significance and application value, but there are some problems. The article mainly described the phenomenon and experimental result, there was not much related discussion of the mechanism, the depth of the article was not enough.

Response: Thanks for the suggestions. We have studied this valuable comment carefully. This work focused on the phenomenon and experimental result of volatile compounds and antioxidant capacity of cat food attractant produced by Maillard reaction, in order to explore a potential and effective way of the application of Maillard reaction in cat food attractant processing. Presently, we are studying on bioactive properties of Maillard reaction products generated from hydrolysate of grass carp waste reacted with glucose and cysteine hydrochloride, including the bioactivities of volatile MRPs and the antioxidative activity of MRPs. In our future work, we will analyze the functional properties of MRPs derived from grass carp waste with glucose and cysteine hydrochloride. Therefore, in this study we did not show too much about the mechanism of experimental, we will descript the details of the formation of volatile substances and antioxidative activity of MRPs clearly in our future study.

  1. The introduction is not concise enough and it should be refined.

Response: Thank you for your comment. We have revised it in the revised manuscript according to the comments.

  1. As compared with the commercial cat-food attractant, the self-made cat-food attractant has weaknesses in first smell, first bite, and feeding rate, so I think the formulation of cat-food attractants needs to be further improved.

Response: Thank you for carefully reading our manuscript and giving us suggestion. We will add some umami ingredients, flavors, and spices in our self-made cat-food attractant refer to the commercial formula, then analyze the sensory properties, acceptance and palatability of self-made cat-food attractant.

  1. The qualitative and quantitative methods of volatile substances are not accurate, only spectrum library was used for qualitative and peak area normalization method was used for semi-quantification in this study.

Response: Thanks. We have studied this valuable comment carefully. In this study, the mass spectrum was matched with MEANLIB, REPLIB, NISTDEMO, and Wiley Library for qualitative of match retrieval, and the matching degree of more than 800 (maximum 1000) was used as basis for identification. We found that some studies evaluated the volatile substances have used the similar methods [1-4], and the matching degree of our results were more than 800 (maximum 1000), so in our opinion, spectrum library was enough for this study. In the future, for more accurate results, we could use gas chromatography-ion mobility spectrometry (GC-IMS) to analyze the volatile substances of our products. The peak area normalization method was used for semi-quantification in this study, which was not a very precise method. The purpose of this study was mainly focus on the difference between commercial cat-food attractant and self-made cat-food attractant, found out the drawback of self-made attractant, then we will add some umami ingredients, flavors, and spices in our self-made cat-food attractant to improve the flavor of our products. So next step, we will produce better products through parameter optimization, the content of volatile substances will be quantified by internal standard method.

  1. Domínguez, R.; Purriños, L.; Pérez-Santaescolástica, C.; Pateiro, M.; Barba, F. J.; Tomasevic, I.; Campagnol, P. C. B.; Lorenzo, J. M., Characterization of Volatile Compounds of Dry-Cured Meat Products Using HS-SPME-GC/MS Technique. Food Analytical Methods 2019, 12, (6), 1263-1284.
  2. Zhu, C. Z.; Zhao, J. L.; Tian, W.; Liu, Y. X.; Li, M. Y.; Zhao, G. M., Contribution of Histidine and Lysine to the Generation of Volatile Compounds in Jinhua Ham Exposed to Ripening Conditions Via Maillard Reaction. J Food Sci 2018, 83, (1), 46-52.
  3. Petricevic, S.; Marusic Radovcic, N.; Lukic, K.; Listes, E.; Medic, H., Differentiation of dry-cured hams from different processing methods by means of volatile compounds, physico-chemical and sensory analysis. Meat Sci 2018, 137, 217-227.
  4. Wu, H.; Zhuang, H.; Zhang, Y.; Tang, J.; Yu, X.; Long, M.; Wang, J.; Zhang, J., Influence of partial replacement of NaCl with KCl on profiles of volatile compounds in dry-cured bacon during processing. Food Chem 2015, 172, 391-9.

  1. The composition and content of volatile substances were only analyzed in this study. However, the contribution of volatile substances depends not only on their composition and content, but also in combination with other methods such as sniffing to identify the key volatile substances.

Response: Thank you for carefully reading our manuscript and giving us suggestion. We know that only GC-MS could not fully characterize the contribution of volatile substances. There are other methods such as electronic nose, gas chromatography olfactometry (GC-O), gas chromatography-ion mobility spectrometry (GC-IMS) could be used to identify the key volatile substances. So, in our future study, we will use a combination of equipment to analyze the key volatile substances.

  1. In Table 5, the format of the content needs to be modified

Response: Thank you for your comment. We have revised it in the revised manuscript according to the comments.

  1. P341, change “Scheme 3.” to “Group 3:”. Please be consistent.

Response: Thank you for your comment. We have checked the manuscript but did find Scheme 3 in P341, I think it's a typographical problem. I have made modification instructions in the revised draft.

  1. In Table 7, groups A and B both stand for the commercial attractants? Also, in table 8, what do groups A and B represent for?

Response: Thanks for the suggestion. This is an error in original manuscript caused by our carelessness. Group A: Blank cat food + 3% self-made attractants. Group B: Blank cat food + 3% commercial attractants. Group CK: Blank cat food. And we have checked the rest of the tables, including table 8.

  1. Please keep the reference format consistent.

Response: Thank you for your comment. We have revised it in the revised manuscript according to the comments.

We really appreciate the Editor and Reviewers’ work, and we hope the revised manuscript will meet the requirements of publication.

Yours sincerely,

Ji Luo, Ph. D

College of Life Science, Anhui Normal University.

Reviewer 3 Report

1- page 2 line 95-96 , It is necessary to transfer these results to the final conclusion of the research

2- page 2 line 98, It is necessary to separate the preparation of materials from chemicals, each writing separately, that is better for the reader

3- the References need updating

4- page 3 line 111, Did the researcher validate the method after making the modification to it?

5- page 3 line 133, The researcher should add the meanings of all the abbreviations under the equation

6- I suggest deleting the first and second figures and sufficing with the tables (second to sixth)

7- In all tables (second to sixth) what do the written letters (a,b,c,d &e) mean and must be written under the table?

Author Response

RESPONSE TO REVIEWER’S COMMENTS

Ref. No.: molecules-1965662

Title: Effects of Maillard reaction on volatile compounds and antioxidant capacity of cat food attractant

  Thank you for your comments concerning our manuscript (molecules-1965662), and those comments are all valuable and very helpful for revising and improving our paper. We have studied the comments carefully and have made corrections which we hope that will meet with approval. The changes in the revised manuscript have been marked in “Colored Text”. Here are the detailed responses addressing point to point according to the order of the comments.

Reviewer: 3

  1. page 2 line 95-96, It is necessary to transfer these results to the final conclusion of the research.

Response: Thanks for your professional suggestion and the recommended paper. We have revised it in the revised manuscript according to the comments.

  1. page 2 line 98, It is necessary to separate the preparation of materials from chemicals, each writing separately, that is better for the reader.

Response: Thanks for your professional suggestion and the recommended paper. We have separate the preparation of materials from chemicals.

  1. the References need updating.

Response: Thank you for your comment. We have updated the references.

  1. page 3 line 111, Did the researcher validate the method after making the modification to it?

Response: Thank you for your comment. The method of preparing protein hydrolysate from grass carp waste we used in this study was based on our preliminary experiments, and we have validated the method after making the modification to it. The complex of trypsin and flavor protease was used to deeply hydrolyze grass offal, response surface methodology (RSM) was applied to optimize the enzymatic hydrolysis process of grass offal protein and maximize the content of free amino acids in its hydrolysates by assessing the factors of enzymatic hydrolysis temperature, pH, liquid material ratio and enzyme addition, using total free amino acids as response factor.

  1. page 3 line 133, The researcher should add the meanings of all the abbreviations under the equation.

Response: Thanks for the suggestion. We have included all the abbreviations in the paper, so they are not marked in the equation, we think it is clearer to make consistent marks in the paper.

  1. I suggest deleting the first and second figures and sufficing with the tables (second to sixth)

Response: Thank you for your suggestion. For the Fig 1 and 2, we express the composition and relative content of volatile compounds with figures, in order to state the results of our experiments and let readers have a more intuitive impact, so we think the figures will help the reader to distinguish the flavor difference between the two attractants more quickly, since the substances in the table 2-6 are more specific and more time consuming. So we thought it would be better to leave the figures.

  1. In all tables (second to sixth) what do the written letters (a,b,c,d &e) mean and must be written under the table?

Response: Thank you for your comment. Data with different letters in the same row (a, b, c, d & e) were significantly different (P < 0.05). I think it's a typographical problem, not putting the instructions under the table. I have made modification instructions in the revised draft.

We really appreciate the Editor and Reviewers’ work, and we hope the revised manuscript will meet the requirements of publication.

Yours sincerely,

Ji Luo, Ph. D

College of Life Science, Anhui Normal University.

Round 2

Reviewer 2 Report

The response to reviewers is acceptable, and it is recommended to receive this article.

Reviewer 3 Report

non